# Understanding of Final Year Medical, Pharmacy and Nursing Students in Pakistan towards Antibiotic Use, Antimicrobial Resistance and Stewardship: Findings and Implications

**DOI:** 10.3390/antibiotics12010135

**Published:** 2023-01-10

**Authors:** Iqra Raees, Hafiz Muhammad Atif, Sabahat Aslam, Zia Ul Mustafa, Johanna Catharina Meyer, Khezar Hayat, Muhammad Salman, Brian Godman

**Affiliations:** 1Department of Medicine, Faisalabad Medical University, Faisalabad 38000, Pakistan; 2Department of Medicine, Rural Health Centre (RHC), Kamer Mushani, Mianwali 42200, Pakistan; 3Department of Medicine, University Medical and Dental College, Faisalabad 38000, Pakistan; 4Discipline of Clinical Pharmacy, School of Pharmaceutical Sciences, University Sains Malaysia, Gelugor 11800, Penang, Malaysia; 5Department of Pharmacy Services, District Headquarter (DHQ) Hospital, Pakpattan 57400, Pakistan; 6Department of Public Health Pharmacy and Management, School of Pharmacy, Sefako Makgatho Health Sciences University, Ga-Rankuwa 0208, South Africa; 7Institute of Pharmaceutical Sciences, University of Veterinary and Animal Sciences, Lahore 54000, Pakistan; 8Institute of Pharmacy, Faculty of Pharmaceutical and Allied Health Sciences, Lahore College for Women University, Lahore 54000, Pakistan; 9Centre of Medical and Bio-Allied Health Sciences Research, Ajman University, Ajman P.O. Box 346, United Arab Emirates; 10Strathclyde Institute of Pharmacy and Biomedical Science (SIPBS), University of Strathclyde, Glasgow G4 0RE, UK

**Keywords:** antimicrobial resistance, antibiotic use, education, knowledge, medical, pharmacy, nursing final year students, Pakistan

## Abstract

Antimicrobial resistance (AMR) is a leading public health threat, which is exacerbated by the high and inappropriate use of antibiotics. Consequently, there is a need to evaluate knowledge regarding antibiotic use, AMR and the readiness to implement antimicrobial stewardship programs (ASPs) among final year medical, pharmacy and nursing students in Pakistan. This reflects the high and increasing rates of AMR in the country, and students as future healthcare professionals (HCPs). A cross-sectional study was conducted among 1251 final year students from 23 public and private educational institutions in Punjab. The majority of the surveyed participants possessed good knowledge of antibiotic use, AMR and the potential causes of AMR. The most common sources of the information on antibiotics were smartphones (69.9%), peers (35.9%) and medical textbooks (30.6%). However, most surveyed participants were not fully prepared to participate in ASPs. They knew, though, how to reduce AMR by educating HCPs about appropriate prescribing, implementing ASPs and improving laboratory facilities. There was a significant association between antibiotic knowledge and causes of AMR with sex, family income and student type (*p* < 0.05). Being a student at a public sector university (OR = 4.809; CI = 3.261–7.094; *p* < 0.001) and age (OR = 0.524, CI = 0.327–0.842; *p* < 0.008) were among the key factors impacting students’ training on ASPs. Educational curricula must be improved to include more information about appropriate antibiotic use and ASPs, along with sufficient training, workshops and clinical rotations in the final year, to fully equip students by graduation.

## 1. Introduction

Antimicrobial resistance (AMR) is one of the principal global public health threats, challenging healthcare delivery systems across countries. This places an additional economic burden on countries, in addition to increasing morbidity and mortality [1,2,3,4]. Nearly 5 million (4.95) deaths globally were associated with AMR in 2019, with 1.27 million deaths directly attributable to bacterial AMR [1]. According to the Centers for Disease Control and Prevention (CDC), AMR is responsible for the loss of USD 55 billion every year in the United States of America [5]. If no actions are taken to improve the appropriateness of antimicrobial prescribing and thereby reduce AMR, AMR could potentially result in an economic loss of more than 5% of GDP and 10 million deaths annually by 2050 [3,6,7,8]. These concerns have resulted in multiple international and national activities to address the crisis. Actions include the development of the World Health Organization’s (WHO) Global Action Plan on AMR and resultant National Action Plans (NAP), including Pakistan’s NAP [9,10,11,12,13].

Whilst AMR affects every country, the greatest burden of AMR is among low- and middle-income countries (LMICs) [1,14]. This arises from numerous political, economic, socio-cultural and environmental factors among LMICs [15,16,17]. These include poor surveillance regarding antibiotic use patterns, exacerbated by a lack of electronic monitoring systems; inappropriate hygiene practices, including a lack of regular hand washing; a lack of awareness in the general population about the risks of AMR; limited diagnostic facilities to differentiate bacterial from viral infections; irrational prescribing, including the extensive prescribing and dispensing of antibiotics for self-limiting conditions; insufficient staff capacity and their training, as well as communication gaps among healthcare providers [12,16,17,18,19,20,21,22]. Alongside this, the prevalence of infectious disease outbreaks, including malaria, dengue fever, typhoid, tuberculosis and Human Immunodeficiency Virus/Acquired Immunodeficiency Syndrome (HIV/AIDS), places an additional burden on the poorly developed health infrastructures in LMICs, including Pakistan [23,24]. These additional infectious diseases, and their impact on available resources, further compromise initiatives to improve the rational use of antimicrobials among LMICs. These challenges have been further exacerbated by the recent COVID-19 pandemic [25,26]. Most of the healthcare facilities in many LMICs, especially in rural areas, currently lack qualified and trained healthcare workers (HCWs). This leads to inappropriate medicine use, including antimicrobials, resulting in the need to rapidly implement agreed NAPs [12,13,17]. There are also concerns regarding the inappropriate prescribing of antimicrobials in hospitals, further exacerbated by the COVID-19 pandemic, despite few bacterial or fungal co-infections (3.5–14.3% of patients) [20,27,28,29,30,31].

Pakistan is the third-largest consumer of antimicrobials among LMICs, followed by India and China, with a 67% increase in their use from 2000 to 2015, which is similar to other South Asian countries [32,33]. In addition, an appreciable proportion of the global burden of AMR is reported to be from South Asian countries, including Pakistan [34]. Both multi-drug-resistant (MDR) and extensively drug-resistant (XDR) bacteria have been identified in different regions of Pakistan, representing an appreciable problem globally due to associated economic costs and mortality with rising AMR rates [35,36,37]. A previous meta-analysis on the extent of AMR in Pakistan indicated that 90% *Salmonella* spp. were resistant to ciprofloxacin, more than 70% of *Pseudomonas* spp. were resistant to ceftazidime and 96% of *Enterococcus* spp. were resistant to erythromycin [34]. Moreover, previous studies from Pakistan have highlighted poor compliance with hand hygiene protocols in hospitals [38], poor adherence to standard treatment guidelines [39,40,41], overprescribing of antibiotics where these are not needed, overprescribing of broad-spectrum antibiotics instead of narrow-spectrum ones among different age groups and the extensive self-purchasing of antibiotics without a prescription, combined with poor enforcement of drug regulations [42,43,44,45,46,47,48,49,50]. All of these practices need to be urgently addressed to reduce future AMR.

In compliance with WHO recommendations, Pakistan developed its own NAP in 2017, advocating for the implementation of key activities. These included enhancing the national awareness of AMR and implementing behavioral change strategies to reduce AMR, estimating current antimicrobial utilization patterns alongside the health and economic burden of AMR, as well as the inclusion of AMR in all public health research activities [13,51]. However, there are challenges in the implementation of the NAP in Pakistan. Challenges include the lack of a robust healthcare system and policies to ensure rational antimicrobial use, as well as available resources, including personnel, to fully implement the NAP [52].

Healthcare professionals (HCPs) play vital roles in the prescribing, dispensing and administration of antibiotics, as well as educating other HCWs about the causes of AMR and how to prevent it, especially in LMICs [53]. Previous studies among HCPs in the different regions of Pakistan have indicated that they possess insufficient training and knowledge about antibiotic use, resistance and hospital-based antimicrobial stewardship programs (ASPs) [54,55,56,57,58,59,60,61,62,63,64]. This is important as these studies highlight that hospital or community-based AMS programs are essential to improve patient care as well as address AMR. Nevertheless, the medication use process, from prescribing to administration and monitoring, is highly complex, necessitating interprofessional coordination among HCPs, including doctors, pharmacists and nurses.

The appropriate clinical care of patients depends upon the knowledge and attitudes of the HCWs, coupled with the implementation of certain standards of care to ensure safe and responsible practices [65,66,67]. Moreover, the WHO has stressed that awareness, training, appropriate use, diagnostic stewardship and surveillance, as well as coordination among HCWs, is necessary to combat rising AMR rates [68]. To date, only a few studies have been conducted among medical and pharmacy students in Pakistan about their awareness concerning antibiotic use, AMR and ASPs [69,70]. The current study aimed to address this by evaluating the knowledge of antibiotic use and AMR, and the preparedness towards ASPs, among final year medical, pharmacy and nursing students from Punjab Province in Pakistan. Alongside this, we sought to determine the most significant factors that could impact ASP training among students. We built on earlier studies to provide a broad range of student HCPs in a single study with the growing role of nurses as part of ASPs and Infection, Prevention and Control (IPC) groups within hospitals in LMICs, as well as with medication use reviews [12,71]. The findings can be used to guide improvements in educational curricula to fully equip students by graduation.

## 2. Results

A total of 1472 final year medical, pharmacy and nursing students from 23 educational institutions (9 medical, 7 pharmacy and 7 nursing) were invited by the investigators to take part in the survey. Out of these, 1251 finally participated, giving a response rate of 84.9%. Among the responders, 471 (37.6%) were pharmacy students, 420 (33.6%) were nursing students and 360 (28.8%) were medical students (Appendix A). Most study participants were female (73.2%, *n* = 916), aged between 20 and 25 years (88.3%, *n* = 1105) and residing in rural areas (60.1%, *n* = 752). As far as the professions of the parents of the study participants were concerned, the majority (81.1%, *n* = 1014) were in non-medical professions. Only 29.5% (*n* = 369) of the study participants had received ASP-related training.

Overall, 78.1% (*n* = 977) of those surveyed knew that antibiotics may cause allergic reactions, 81.8% (*n*=1023) knew that they are not useful in treating viral infections, and a similar proportion were aware of antibiotic resistance (87.7%, *n* = 1097) (Table 1. However, 43.5% (*n* = 544) did not know that antibiotics may cause secondary infections, and 60.4% (*n* = 756) knew that skipping one or more than one dose can lead to resistance development. Meanwhile, 43.3% (*n* = 541) were unaware of key issues regarding cross-resistance and 47.3% (*n* = 591) did not know the appropriate indications for antibiotic use. In addition, most of the study population had never been taught about AMR (76.8%, *n* = 961) and ASPs (91.7%, *n* = 1147) in their current curriculum (Table 1).

Awareness of the potential causes of AMR among the study participants is summarized in Table 2. Overall, 96.1% (*n* = 1203) of the study participants strongly agreed or agreed that the overprescribing of antibiotics is a principal cause of AMR. Meanwhile, 85.7% (*n* = 1072) knew that the prescribing of broad-spectrum antibiotics was a key cause of AMR, along with a long duration of antibacterial therapy (73.2%, *n* = 916). Other potential causes of AMR identified by the study participants were inadequate IPC measures among HCPs (93.2%, *n* = 1166) and the sale of antibiotics without a prescription at pharmacies (66.7% strongly agree and 19.5% agree, *n* = 1078). However, 48.1% (*n* = 592) disagreed or strongly disagreed that excessive antibiotic use in livestock was a cause of AMR.

The perceptions about antibiotic use and resistance among study participants are shown in Table 3. Overall, 77.5% (*n* = 970) of the study participants strongly agreed that AMR is a global health issue, including within Pakistan (82.3%, *n* = 1030). Meanwhile, 77.6% (*n* = 971) of the study participants also strongly agreed that good knowledge regarding antibiotics and their use was important for their future professional career (77.6%, *n* = 971), that the irrational use of antibiotics was professionally unethical (66.5%, *n* = 832) and that the inappropriate use of antibiotics can cause harm to patients (75.5%, *n* = 945) (Table 3).

Commonly used resources that were referred to by the study participants for the purpose of learning about the appropriate use of antibiotics and AMR included textbooks, infectious and non-infectious disease physicians, peers and internet resources, which included smartphone applications as well as Pharmapedia (Appendix A). However, 89.8% (*n* = 1124) seldom or never referred to medical journals for information and 92.2% (*n* = 1153) of the study participants had never referred to guidelines such as those from the Infectious Diseases Society of America (IDSA). In addition, 87.2% (*n* = 1091) rarely (87.2%) consulted the Sanford guide for antibacterial treatment guidance (Appendix A).

Perceptions regarding the preparedness of the study participants for undertaking ASPs are shown in Table 4. Overall, 36.5% (*n* = 456) of the study participants stated that they were poorly prepared for making/interpreting diagnoses, only 34.1% (*n* = 427) knew when to start/stop antibiotics (very good or good), and 44.1% (*n* = 552) believed that they were poorly prepared in terms of the selection of appropriate antibiotics. Poor or very poor perceptions of their preparedness regarding antibiotic dose calculations, transitioning from intravenous to oral routes of antibiotic administration and an understanding of the antibiotic spectrum and its significance, including interpreting antibiograms, were also reported among 61.0.7% (*n* = 763), 4=53.1% (*n* = 664), 64.2.0% (*n* = 803)and 50.9% (*n* = 637) of study participants, respectively.

HCPs’ education, the implementation of ASPs, appropriate diagnostic facilities and the development of institutional treatment guidelines were seen as common ways to reduce AMR among the study participants, with high rates strong agreement and agreement (Table 5). Other measures to reduce AMR that were identified by the study participants included discouraging antibiotic prescription over the telephone, as well as self-medication of antibiotics without a prescription.

The associations between the demographic characteristics of the study participants and their knowledge of antibiotics, awareness about causes of AMR, perceptions of AMR, preparedness towards ASPs and approaches to tackle AMR are documented in Table 6. A statistically significant association was identified between knowledge of antibiotics and the causes of AMR with sex, family income and student type (*p* < 0.05). The perceptions of students towards AMR and their preparedness regarding implementing ASPs were significantly associated with sex, student type, institute type and residence (*p* < 0.05). The approaches used to tackle AMR had a significant association with only student type and institute type (*p* < 0.05). The detailed results are contained in Table 6.

We also investigated the factors that could impact the ASP training of students. Being a student from a public sector university (OR = 4.809, CI = 3.261–7.094; *p* < 0.001), their age (OR = 0.524, CI = 0.327–0.842; *p* < 0.008) and the category of students, including a medical (OR = 0.177, CI = 0.110–0.285; *p* < 0.001) or pharmacy student (OR = 0.206, CI = 0.125–0.341; *p* < 0.001), are the most likely factors that could influence the students’ training regarding ASPs (*p* < 0.05).

## 3. Discussion

We believe that this is the first paper in Pakistan that has evaluated the different perspectives of antibiotic use, AMR and ASPs among future medical doctors, pharmacists and nurses among both public and private sector institutions in Pakistan. We are aware that studies have been published with different student cohorts across LMICs including Pakistan; however, we are unaware of any studies that have combined all three key student populations among both public and private universities in LMICs [69,70,72,73,74,75,76,77,78,79,80,81,82,83]. This is increasingly important, with multidisciplinary teams being key to improving future antibiotic prescribing as part of ASPs [27].

Encouragingly, the majority of the study participants had appropriate knowledge about antibiotics and their use. This included their lack of effect on viral infections, potential allergic reactions associated with antibiotic use and the importance of AMR. These findings regarding their knowledge of antibiotics and their use are similar to previous studies conducted among LMICs, as well as those among nursing students in Spain [72,74,75,82,84,85]. However, our findings are different from other studies in LMICs, where there have been concerns regarding the knowledge of HCP students with respect to antibiotics and AMR [65,73,80,86,87,88,89]. Good knowledge regarding antibiotics, AMR and ASPs is important to help improve future utilization rates and reduce AMR, particularly in LMICs. In the case of pharmacy students, we are aware of studies conducted among African as well as among Central and Eastern European countries indicating that the presence of well-trained pharmacists appreciably limits the dispensing of antibiotics without a prescription, with alternative treatments suggested [22,90,91,92]. Advice from community pharmacists regarding alternative treatments, especially for self-limiting infections, is enhanced by the availability of robust guidelines [92].

Most of the study participants stated that the potential causes of AMR in Pakistan were excessive antibiotic prescribing, particularly broad-spectrum antibiotics; poor compliance with infection control measures among HCPs, especially in hospitals; and the availability of antibiotics at pharmacies without valid prescriptions. This is also encouraging. These findings are similar to a previous study conducted among medical students in Nigeria, which highlighted that the sale of antibiotics without a prescription, coupled with poor infection prevention measures among HCPs, was associated with increasing AMR [93].

Encouragingly, in addition, most of the study participants considered AMR primarily as a global health crisis, which had occurred due to excessive antibiotic use. Moreover, they also stated that inappropriate antibiotic use was unethical and could cause harm to patients; consequently, appreciable knowledge of antibiotics is essential for student HCPs post-qualification. These findings were in contrast to recent studies from Saudi Arabia, where more than half of the studied pharmacy students were unaware that AMR is a health threat, and in Rwanda, where 37.6% of HCP students did not believe that antibiotics were being overused [80,94]. However, 83% of students in Rwanda were unfamiliar with the concept of antimicrobial stewardship and 49% had not discussed AMR as part of their education [80].

Textbooks, infectious- and non-infectious-disease physicians and smartphone applications were common sources of information among the study participants. This is encouraging as previous studies have shown in LMICs that pharmaceutical companies can be a principal source of information on antibiotics [95,96]. This is a challenge, especially in LMICs, where there are no enforced regulations influencing the objectiveness of information provided by pharmaceutical companies. This can be problematic in terms of the often limited information that is provided by companies, especially surrounding adverse reactions and safety issues, and with companies seeking to maximize the sales of their medicines [97,98,99,100]. This can be counterproductive in this situation, with the goal of governments to improve the rational use of antibiotics as part of NAPs [12,13]. Of equal concern is that international guidelines and journals were infrequently used by study participants to obtain information about antibiotic use and AMR. However, these findings are similar to a previous systematic review that reported that most medical students did not typically use international guidelines and journals to obtain the latest information about antibiotics and AMR [86]. Guidelines among LMICs need to be readily available, easy to use and regularly updated to enhance their use in practice [101,102]. Prescribing guidance is also enhanced across LMICs with the recent availability of the WHO AWaRe (Access, Watch, Reserve) antibiotic book [103].

Another concern highlighted by our study is that most of the participants believed that they were not adequately prepared to enhance rational and appropriate antibiotic use under the umbrella of ASPs. This was because they believed that they were not fully prepared to interpret appropriate microbiological results, when to start/stop antibiotics with correct doses and the concept of ASPs generally. There have been similar concerns regarding AMR and ASP awareness among students in African countries, as well as among HCPs in a number of LMICs [65,87,88,104,105]. This is an issue that needs to be urgently addressed given increasing concerns regarding the high rates of the inappropriate prescribing of antimicrobials and AMR in Pakistan [13,48,52,55,106]. These concerns may be due to a lack of information regarding ASPs and their importance in the current curriculum, which needs to be urgently addressed. We are aware that there have been concerns that it is difficult to fully instigate ASPs in LMICs due to resource and available personnel issues, as well as a lack of knowledge [41,107]. However, this is changing, as seen, for instance, in Africa, as well as other LMICs [22,27,108,109,110].

The reasons for the differences regarding the preparedness to undertake ASPs between the different student types and their ages are unclear, with no impact of other factors, including their gender, place of residence or family profession. These differences have been seen in other studies among LMICs [65,75]. This, however, may reflect differences in curricula between the different schools, as well as greater knowledge gained during training, with typically more training on antibiotics, AMR and ASPs in the later years, along with greater senior-level input [65,67,111]. Differences between the different university types may also reflect the greater academic staff to student ratios among private universities in Pakistan [112].

We are aware of a number of limitations with our study. Firstly, we only collected data from final year medical, pharmacy and nursing students among both public and private sector training institutions in Punjab Province; consequently, we cannot generalize our results to the rest of the country due to under-coverage bias as a result of the non-probability sampling. Moreover, typical inference through regression modeling is also one of the limitations of our study. However, these shortcomings can be addressed in future projects by recruiting study participants though different sampling techniques. We also did not explore the reasons for the differences regarding ASPs and their implementation between the different student types. Despite these limitations, we believe that our findings have provided appropriate insights for public health experts, clinicians, academicians, teachers and policymakers to identify pertinent knowledge gaps to address in future training programs. This is based on the extensive methodology that we used when developing and testing the questionnaire and its content. In view of this, we believe that our findings are robust can be used to build initiatives among future HCPs to enhance rational antibiotic use and reduce AMR in the future in Pakistan. We will be following this up in future studies.

## 4. Materials and Methods

### 4.1. Study Design, Population and Location

A cross-sectional study design was employed in this survey. Data were subsequently collected from final year medical, pharmacy and nursing students currently enrolled among 23 public and private sector educational institutions in seven divisions (out of a total of ten) in Punjab Province. As per the information from the Pakistan Medical and Dental Association (PMDC) and Pakistan Medical Commission (PMC), there are currently 19 public sector and 35 private sector medical colleges/universities operational in Punjab, offering Bachelor’s degrees in medicine (five years plus one year residency). Each institution has between 50 and 100 students in their final year [113]. There are currently seven public sector and eight private sector universities offering a Doctor of Pharmacy (five years) graduation program in Punjab Province, based on information provided by a representative from the Pakistan Pharmacy Council. Each institution currently has between 50 and 150 final year pharmacy students. According to the Pakistan Nursing Council (PNC), there are 51 public sector and 17 private sector nursing institutes offering a graduate program in nursing (four years), and each institution has between 50 and 80 final year nursing students [114].

The final sample of institutions, based on convenience sampling, included nine medical, seven pharmacy and seven nursing colleges/universities from both the public and private sectors (Appendix A). This was seen as the most appropriate approach for this study.

### 4.2. Study Instrument

We used a previously validated study instrument that had been employed in recent studies undertaken in Pakistan. This was part of a study project entitled ‘Antibiotic use, resistance and stewardship among HCWs and students’ [69,70]. These studies were conducted among medical and pharmacy students in different cities of Punjab. The objective was to ascertain their awareness, perceptions and key issues surrounding antibiotic use, resistance and preparedness towards ASPs. The internal consistency, or reliability, of the study instrument was also measured by using Cronbach’s alpha. This showed a value greater than 0.7, suggesting reasonably strong internal consistency.

A pilot study was initially conducted among thirty medical, pharmacy and nursing students to again measure the reliability of the instrument (20 participants from each category). The overall value of the study instrument fell within the acceptable range of internal consistency. In addition, the students stated that they fully understood the content of the instrument. After considering the findings and recommendations of the pilot study participants, which principally involved details of demographic variables, the final version of the study instrument incorporated the following seven sections:**Section I:** Contained ten items related to the demographic details of the study population, including their age, gender, institution type, the profession of their parents and any previous training/research experience related to antibiotic utilization, AMR and ASPs.**Section II:** Consisted of eleven questions relating to the knowledge of antibiotics and AMR. Each question had three options: ‘yes’, ‘no’ and ‘don’t know’.**Section III:** Consisted of ten questions to extract information about the potential causes of AMR. Study participants were requested to select one option on a 5-item Likert scale, with the 5 items ranging from strongly agree to strongly disagree, as typically seen in such scales [83,115].**Section IV:** Dealt with the perceptions of the study population regarding antibiotic utilization and AMR. This section contained nine questions with a 5-item Likert scale ranging from strongly agree to strongly disagree.**Section V:** Collected information about the common sources of information among the study participants concerning antibiotic utilization, AMR and ASPs.**Section VI:** Consisted of ten questions about the perceptions of the preparedness of the study participants concerning ASPs and their implementation. Each question contained five options, similar to other sections.**Section VII:** Contained ten questions relating to the different means of combating AMR. Study participants were again requested to select one option on a 5-item Likert scale, from strongly agree to strongly disagree.

### 4.3. Sampling Technique and Size Calculation

A convenience sampling technique was employed for this survey. The sample size was calculated using the Raosoft 206-525-4025 (http://www.raosoft.com/samplesize.html, accessed on 24 October 2022) online sample size calculator. Assuming a population size of 5000 for each of the three student categories, i.e., medical, pharmacy and nursing students, and an expected response distribution of 50% in case of the largest sample, at a 95% confidence level and 5% margin of error, the minimum sample size for each student category was calculated as 376, giving an overall minimum sample size of 1128. We increased the sample size by 30% to ±1466 to account for any incomplete data or non-responders, to enhance the robustness of the findings. All public and private sector institutions offering medical, pharmacy and nursing students were contacted by the investigator for participation in the current survey. Only those institutions whose administration/societies allowed or were supportive of this study were included in the survey.

### 4.4. Data Collection Procedure

The questionnaire was created as an online Google Form (https://docs.google.com/forms/ accessed on 16 May 2022) comprising the seven different sections, active during the data collection time period. Various social media sites were used, including “WhatsApp”, “Facebook Messenger” and “Gmail”, and Google Form links were used to conduct the survey depending on the preferences and convenience of the participants with the assistance of institutional administration/societies. Participation in our survey was voluntary. Written informed consent was provided by all the participants prior to their enrollment in the study.

The online link to the data collection form was subsequently shared on the above-mentioned social media platforms, as well as among private and professional/institutional groups/societies. To encourage participation, participants were given the option of answering all of the questions by simply clicking on a link. Those who were unwilling to participate could simply click on the given link and go no further. All of the data were collected through the online Google Form. The data were subsequently exported to MS Excel and then into a data analysis software system for analysis.

### 4.5. Inclusion and Exclusion Criteria

All final year students among targeted medical, pharmacy and nursing public and private educational institutions in Punjab Province, and those who were willing to participate in our survey, were included. Students who were (i) not studying medicine, pharmacy or nursing in Punjab; (ii) not in their final year of study; and (iii) from institutions outside of Punjab Province were excluded from this study.

### 4.6. Statistical Analysis

All data analysis was performed using the Statistical Package for the Social Sciences (SPSS Inc., version 18, IBM, Chicago, IL, USA). Descriptive statistics, including frequencies and percentages, were used to analyze the data. Moreover, the normality of the data was checked through Shapiro–Wilk and Kolmogorov–Smirnov tests. Median scores were calculated for each domain. Moreover, the independent-samples Mann–Whitney U test and independent-samples Kruskal–Wallis test were performed to test if there were differences among characteristics of the students with regard to the knowledge, causes, perceptions, preparedness and approaches used to address AMR. *p*-values < 0.05 were considered statistically significant. Using multivariate regression analysis, we also investigated the factors that could impact the ASP training of the students.

### 4.7. Ethical Considerations

Ethical approval for the current study was obtained from the Human Research Ethics Committee, Department of Pharmacy Practice, the University of Lahore (REC/DPP/FOP/UOL/64). Furthermore, permission to conduct the study was also obtained from the office of the institutional head of all participants’ institutions before data collection commenced. A participant consent form was included on the first page of the data collection form. Participants could subsequently exclude themselves at this stage if wished.

All the data collected were kept confidential and no personal information, including names, addresses and cell phone numbers, was collected from the study participants. All study participants were assigned an anonymized study number, which was kept confidential, for the purpose of verifying the accuracy of recorded data.

## 5. Conclusions and Recommendations

Our survey concluded that final year medical, pharmacy and nursing students did possess appropriate knowledge of antibiotic use and AMR. They were also aware about the potential causes of AMR in Pakistan and globally, and how to address the ongoing AMR crisis. However, they were not sufficiently prepared to take an active role in any ASPs post-qualification. Consequently, a multisectoral approach is recommended to enhance their knowledge and preparedness for such programs.

Educational curricula among future HCPs in Pakistan must now include subjects related to appropriate antibiotic use and ASPs. This includes greater knowledge of the WHO AWaRe classification of antibiotics and the newly launched prescribing guidance, and its significance. Moreover, training, workshops and clinical rotations should help to fill the knowledge gaps among these future HCPs going forward, with input from senior-level personnel providing a good basis for the future. HCP students should also be equipped with international guidelines, in addition to the new WHO AWaRe prescribing guidance for infectious diseases, to improve future prescribing in the absence of robust national or regional guidelines in Pakistan. Alongside this, universities must provide regular access to journals and appropriate information technology sources during their training, which was a concern at the start of the pandemic with the closure of universities. This will help future HCPs to familiarize themselves with the latest information and guidance, ready for ASP activities post-qualification.

## Figures and Tables

**Table 1 antibiotics-12-00135-t001:** Knowledge about antibiotics and antibiotic resistance (*n* = 1251).

Knowledge Statements	Yes (*n*, %)	No (*n*, %)	Don’t Know (*n*, %)
Antibiotics are useful in treating viral infections.	161 (12.9)	1023 (81.8)	67(5.4)
Antibiotics can cause secondary infections by killing normal flora.	707 (56.5)	229 (18.3)	315 (25.2)
Antibiotics can cause allergic reactions.	977 (78.1)	172 (13.7)	102 (8.2)
A resistant bacterium cannot spread in healthcare institutions.	225 (18.0)	859 (68.7)	167 (13.3)
Skipping one or two doses of antibiotics does not contribute to the development of antibiotic resistance	290 (23.2)	756 (60.4)	205 (16.4)
Cross-resistance is the condition in which resistance occurs to a particular antibiotic that often results in resistance to other antibiotics usually from a similar class	710 (56.8)	256 (20.5)	285 (22.8)
Pain and inflammation without any possibility of infection are indications for antimicrobial therapy	160 (12.8)	660 (52.8)	431 (34.5)
Ever heard of antibiotic resistance	1097 (87.7)	127 (10.2)	27 (2.2)
Ever taught antibiotic resistance in your curriculum	179 (14.3)	961 (76.8)	111 (8.9)
Ever heard of antibiotic stewardship	236 (18.9)	849 (67.9)	166 (13.3)
Ever taught about antibiotic stewardship in your curriculum	67 (5.4)	1147 (91.7)	67 (5.4)

**Table 2 antibiotics-12-00135-t002:** Potential causes for antibiotic resistance (*n* = 1251).

Statements on Causes of Antimicrobial Resistance	Strongly Agree (*n*,%)	Agree (*n*,%)	Neutral (*n*,%)	Disagree (*n*, %)	Strongly Disagree (*n*,%)
Too many antibiotic prescriptions	931 (74.4)	272 (21.7)	44 (3.5)	4 (0.3)	0 (0.0)
Too many broad-spectrum antibiotics use	848 (67.8)	224 (17.9)	72 (5.8)	97 (7.8)	10 (0.8)
Too long durations of antibiotic treatment	528 (42.2)	388 (31.0)	157 (12.5)	164 (13.1)	14 (1.1)
Dosing of antibiotics are too low	103 (8.2)	312 (24.9)	234 (18.7)	424 (33.9)	178 (14.2)
Excessive use of antibiotics in livestock	302 (24.1)	34 (2.7)	323 (25.8)	433 (34.6)	159 (12.7)
Not removing the focus of infection (e.g., medical devices or catheters)	462 (36.9)	209 (16.7)	222 (17.7)	292 (23.3)	66 (5.3)
Antibiotic sale without prescription from community pharmacies	835 (66.7)	244 (19.5)	63 (5.0)	93 (7.4)	16 (1.3)
Patient non-compliance with antibiotic treatment	1003(80.2)	157 (12.5)	49 (3.9)	35 (2.8)	7 (0.6)
Poor infection-control practices by healthcare professionals	970 (77.5)	196 (15.7)	47 (3.8)	35 (2.8)	3 (0.2)
Paying too much attention to pharmaceutical manufacturing claims/misleading advertising	524 (41.9)	230 (18.4)	97 (7.8)	285 (22.8)	115 (9.2)

**Table 3 antibiotics-12-00135-t003:** Perceptions about antibiotic use and antibiotic resistance (*n* = 1251).

Statements on Perceptions	Strongly Agree (*n*,%)	Agree (*n*,%)	Neutral (*n*,%)	Disagree (*n*,%)	Strongly Disagree (*n*,%)
Antibiotic resistance is a global issue	970 (77.5)	276 (22.1)	5 (0.4)	0 (0.0)	0 (0.0)
Antibiotic resistance is a serious problem in Pakistan	1030 (82.3)	214 (17.1)	7 (0.6)	0 (0.0)	0 (0.0)
Antibiotics are overused at the hospitals	817 (65.3)	253 (20.2)	92 (7.4)	76 (6.1)	13 (1.0)
Antibiotic resistance is a significant problem at the hospitals	959 (76.7)	229 (18.3)	50 (4.0)	10 (0.8)	3 (0.2)
Strong knowledge of antibiotic is important in career	971 (77.6)	222 (17.7)	58 (4.6)	0 (0.0)	0 (0.0)
Inappropriate use of antibiotics is professionally unethical	832 (66.5)	238 (19.0)	66 (5.3)	95 (7.6)	20 (1.6)
Inappropriate use of antibiotic can harm patients	945 (75.5)	235 (18.8)	60 (4.8)	10 (0.8)	1 (0.1)
Would like more education on antibiotic resistance	1092 (87.3)	159 (12.7)	0 (0.0)	0 (0.0)	0 (0.0)
New antibiotics will be developed in the future that will counter the problem of “resistance”	949 (75.9)	139 (11.1)	104 (8.3)	46 (3.7)	13 (1.0)

**Table 4 antibiotics-12-00135-t004:** Medical, pharmacy and nursing students’ perceptions of preparedness in antimicrobial stewardship (*n* = 1251).

Statements on Preparedness	Very Good (*n*,%)	Good (*n*,%)	Average (*n*,%)	Poor (*n*,%)	Very Poor (*n*,%)
Making/interpreting accurate diagnosis/treatment of infection	155 (12.4)	233 (18.6)	343 (27.4)	456 (36.5)	64 (5.1)
Interpreting pathology and microbiology results	276 (22.1)	257 (20.5)	249 (19.9)	411 (32.9)	58 (4.6)
Knowing when to start/stop antibiotics	204 (16.3)	223 (17.8)	228 (18.2)	496 (39.6)	100 (8.0)
Choosing the correct antibiotic	160 (12.8)	177 (14.1)	270 (21.6)	552 (44.1)	92 (7.4)
Knowledge of dosing/calculations and duration of antibiotics	129 (10.3)	105 (8.4)	254 (20.3)	672 (53.7)	91 (7.3)
How to de-escalate to narrower spectrum antibiotics	31 (2.5)	168 (13.4)	331 (26.5)	510 (40.8)	211 (16.9)
How and when to transition from intravenous to oral antibiotics	73 (5.8)	266 (21.3)	248 (19.8)	558 (44.6)	106 (8.5)
How to interpret antibiograms	35 (2.8)	279 (22.3)	300 (24.0)	433 (34.6)	204 (16.3)
Understanding spectrums of activity of antibiotics	35 (2.8)	158 (12.6)	255 (20.4)	588 (47.0)	215 (17.2)
Understanding basic mechanisms of resistance of antibiotics	102 (8.2)	195 (15.6)	219 (17.5)	619 (49.5)	116 (9.3)

**Table 5 antibiotics-12-00135-t005:** Ways to reduce antibiotic resistance (*n* = 1251).

Statements on Ways to Reduce Antimicrobial Resistance	Strongly Agree (*n*,%)	Agree (*n*,%)	Neutral (*n*,%)	Disagree (*n*,%)	Strongly Disagree (*n*,%)
Educating healthcare professional in terms of appropriate antibiotic prescribing	754 (60.3)	474 (37.9)	22 (1.8)	1 (0.1)	754 (60.3)
Formal teaching on proper usage of antimicrobial agents among healthcare students	808 (64.6)	424 (33.9)	19 (1.5)	0 (0.0)	0 (0.0)
Implementing antibiotic stewardships programs	659 (52.7)	454 (36.3)	132 (10.6)	6 (0.5)	0 (0.0)
Rationalizing antibiotics use in veterinary sector	653 (52.2)	485 (38.8)	87 (7.0)	24 (1.9)	2 (0.2)
Improving diagnostic facilities	801 (64.0)	435 (34.8)	15 (1.2)	0 (0.0)	0 (0.0)
Development of institutional standard treatment guidelines	587 (46.9)	530 (42.4)	133 (10.6)	1 (0.1)	0 (0.0)
Prescribing antibiotics over the phone	1 (0.1)	54 (4.3)	249 (19.9)	534 (42.7)	413 (33.0)
Patient should be advised not to keep part of the antibiotic course for another occasion	480 (38.4)	259 (20.7)	333 (26.6)	142 (11.4)	37 (3.0)
Pharmacists should be discouraged to dispense antibiotics to meet the patients demands	451 (36.1)	344 (27.5)	344 (27.5)	101 (8.1)	11 (0.9)
Self-medication with antibiotics in community should be discouraged	875 (69.9)	241 (19.3)	123 (9.8)	12 (1.0)	0 (0.0)

**Table 6 antibiotics-12-00135-t006:** Associations of demographic variables with antibiotic knowledge, causes and perceptions of AMR, preparedness towards ASPs and approaches to tackle AMR.

Variables	Knowledge of Antibiotics		Causes of AMR		Perceptions of AMR		Preparedness towards ASPs		Approaches to Tackle AMR	
	Mean Rank	*p*-Value	Mean Rank	*p*-Value	Mean Rank	*p*-Value	Mean Rank	*p*-Value	Mean Rank	*p*-Value
**Sex**										
Female	608.33	<0.001	599.31	<0.001	613.89	<0.001	649.48	<0.001	623.33	0.664
Male	674.32		698.99		659.11		561.80		633.30	
**Age**										
<20 years	472.86	0.184	489.96	0.362	584.82	0.908	715.04	0.019	464.46	0.224
20–25 years	624.74		627.81		626.70		615.60		626.50	
>25 years	652.79		625.24		624.53		703.59		638.97	
**Family income (PKR)**										
<25,000	493.25	<0.001	510.61	<0.001	643.00	<0.001	610.83	0.955	636.70	0.148
25,000–75,000	563.21		589.29		586.81		623.96		606.12	
>75,000	701.55		671.81		667.24		628.01		646.27	
**Student type**										
Medical	719.38	<0.001	751.80	<0.001	607.70	<0.001	247.05	<0.001	610.39	0.004
Pharmacy	746.21		638.27		707.96		828.63		668.11	
Nursing	411.15		504.41		549.78		723.58		592.16	
**Institute type**										
Public	702.01	<0.001	630.54	0.463	642.58	0.007	607.37	0.003	640.08	0.023
Private	420.67		613.73		581.21		676.33		587.98	
**Residence**										
Rural	595.82	<0.001	612.07	0.093	603.62	0.007	584.03	<0.001	615.83	0.220
Urban	671.49		647.00		659.73		689.25		641.33	
**Parents’ profession**										
Medical	659.99	0.101	638.00	0.568	629.79	0.856	599.49	0.209	648.88	0.277
Non-medical	618.05		623.19		625.11		632.20		620.65	
**ASP Training**										
Yes	609.73	0.293	661.74	0.023	638.87	0.410	584.15	0.008	634.64	0.583
No	632.81		611.05		620.62		643.51		622.38	
**Antibiotic use in last 6 months**										
Yes	644.74	0.175	670.70	0.001	632.56	0.638	553.97	<0.001	640.20	0.311
No	616.15		602.51		622.55		663.86		618.53	

NB: ASP: Antimicrobial Stewardship, (PKR) = Pakistani Rupee.

## Data Availability

Additional data are available from the corresponding authors on reasonable request.

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
