# Peer review of "Understanding of Final Year Medical, Pharmacy and Nursing Students in Pakistan towards Antibiotic Use, Antimicrobial Resistance and Stewardship: Findings and Implications"

_antibiotics, 2023, doi:10.3390/antibiotics12010135_

Round 1
Reviewer 1 Report
Thank you for the opportunity to review the manuscript, describing an evaluation of the different perspectives of antibiotic use, AMR, and ASP among medical, pharmacy, and nurse students of Pakistan. This study area overlapped with previous studies such as ref.64 and 65 which were investigated by the authors. It is unclear the difference between the purpose and results of this study and those of these studies. There are several concerns to publish this study.
The authors reported a similar questionnaire survey for medical students and pharmacy students, respectively (ref. 64 and 65). This study included medical, pharmacy, and nurse students. The design of this study overlooked that they have another role in the enhancement of antimicrobial stewardship (AMS) and AMR problem.
The mixing of different faculties in the survey makes it difficult to understand the issues and interpret the results. They should be investigated separately, which has already been reported by the authors. This study area and methods partially overlapped with previous studies which were investigated by the authors.
The questionnaire items in this study were different from previous studies, but the questionnaire for specific faculty in the previous studies revealed specific problems in each field. So, the impact and new findings were lost by the mixing of faculties.
The authors should reconsider the study purpose/design and methods to address the specific problems for AMR and AMS.
I think that simply repeating questionnaire surveys will not lead to concrete measures. The authors should design the survey to address the specific countermeasures in Pakistan.
Author Response
Comments and Suggestions for Authors
1) Thank you for the opportunity to review the manuscript, describing an evaluation of the different perspectives of antibiotic use, AMR, and ASP among medical, pharmacy, and nurse students of Pakistan. This study area overlapped with previous studies such as ref.64 and 65 which were investigated by the authors. It is unclear the difference between the purpose and results of this study and those of these studies. There are several concerns to publish this study.
Author response: Thank you. We have included the rationale for the study at the end of the revised Introduction – with previous studies just concentrating either on just Medical or Pharmacy students. In addition, we have tried to resolve all your concerns about our study and hopefully this is now acceptable.
2) The authors reported a similar questionnaire survey for medical students and pharmacy students, respectively (ref. 64 and 65). This study included medical, pharmacy, and nurse students. The design of this study overlooked that they have another role in the enhancement of antimicrobial stewardship (AMS) and AMR problem.
Author response: Thank you. As mentioned in this study we have collected data from all three major groups of future health care workers in Pakistan. Previous studies (Old Reference 64, 65 – new 69,70) were conducted among medical and pharmacy students separately. However, no study has been conducted among nursing students, and combining the HCPs together, which needs to be addressed going forward as nurses are now a key member of ASPs and IPCs in LMICs. We have now upgraded the introduction to reflect this, and hope this is now OK.
3) The mixing of different faculties in the survey makes it difficult to understand the issues and interpret the results. They should be investigated separately, which has already been reported by the authors. This study area and methods partially overlapped with previous studies which were investigated by the authors.
Author response: Thank you. Can we beg to differ. As you are aware from previous comments, tackling AMR needs combined response from all the healthcare professionals. Consequently, we collected data from all three faculties as they are all were involved in medications use process along with ASPs and IPCs in Pakistan - similar to other LMICs. We hope this clarifies the situation.
4) The questionnaire items in this study were different from previous studies, but the questionnaire for specific faculty in the previous studies revealed specific problems in each field. So, the impact and new findings were lost by the mixing of faculties.
Author response: Thank you for this comment. We modified our study instrument (principally demographic variables) following the input and outcomes from the pilot study’s participants from medical, pharmacy and nursing students. Moreover, all three future HCP groups stated they fully understand the content of study instrument. We have mentioned this in methods section under ‘study instrument’. Hope this is now acceptable.
5) The authors should reconsider the study purpose/design and methods to address the specific problems for AMR and AMS.
Author response: Thank you for this. However, before we can discuss ASPs – we need to know current knowledge regarding antibiotics and AMR to then plan appropriate ASPs, etc., which was also part of the questionnaire. We know this from experience of conducting and researching AMR and ASPs across multiple countries and ascertaining the reasons why certain ASPs have been successful. We hope this clarifies the situation.
6) I think that simply repeating questionnaire surveys will not lead to concrete measures. The authors should design the survey to address the specific counter measures in Pakistan.
Author response: Thank you. We have included concrete proposals in the Discussion – as please remember this study was undertaken among students and not practicing HCPs. We hope this is now acceptable.
Reviewer 2 Report
The investigators have completed a valuable study that in essence, indicates the lack of formal and affective education in the curriculum for student HCPs in universities in Pakistan. Generally, the findings are consistent with other studies, in that knowledge and understanding has important gaps, and that understanding action necessary with regards to appropriate prescribing and stewardship, is lacking too.
The study is a basic cross-sectional study. I find the sections on sampling, sampling source, to be confusing, as indicated in my comments. Similarly, the methods for multivariate analysis, adjusting for potential confounders, and primary stated hypotheses, are lacking and/or missing from the study execution. Some of the variables reported did not appear to be used in the analysis, such as income and parental profession - nor was there an explanation as to why this data was collected.
The questionnaire, or at least the representation of the question as written in the tables, has quite a few confusing/ambiguous questions. Whilst the investigators claim to have used a previous questionnaire, that does not mean it is valid and reliable. There is a statement that internal validity was tested, but no evidence of how, nor the specific outcome. The quality of the questionnaire is my biggest concern with then making generalisations from the results.
I found a lot of grammatical errors and sentences that did not make sense. These are indicated below. Some points needed to have more explanation.
The abstract is poorly written and needs to be clearly aligned with the highlights from the results.
61 LMIC needs to be in brackets
63-67 needs some data - it's all a bit vague.
67-71 - need to either make a brief statement, and nothing else, about added burden on an already weak healthcare service; or you need to expand and epxlain the points
72 - give data on the level of inappropriate use of antibiotics
76 - give data
81 - why an appreciable problem globally? explain
87 - 'these' instead of 'this'
88 - I would make the point somewhere about the regulation and over-the-counter purchasing
95 - give examples of the challenges
51 - I don't think the role of HCPs is limited to LMIC - it is a global need
100 - give an example of what they don't know. Also is this comparable to other countries or unique to Pakistan?
101 - This point has already been made. Delete sentence.
105 - I would start a new paragraph that makes the point that the basis for appropriate clinical care starts from the beginning of their training
Then add that this is a concern, and give a few more examples - including a recent study by David McMaster et. al., doi: 10.1093/jacamr/dlaa096. eCollection 2020 Dec.
and also work by Molly Courtenay
110 - how could you 'invite' the participants if you had simply used social media as a way to try and recruit anyone seeing the social media post?
and how could you determine the number initially invited?
117 - how did you define 'rural area'?
117 29.5% is not a quarter
117 the point about ASP training should come after all the demographics were explored.
119 I don't understand why you collected data on parent profession
125 - 'the majority....' this is subjective, and should be in the discussion. Results should stick to the facts.
Table 2 - skipping doses - I suspect it needs to be more than one or two doses.
Cross-resistance - the question is confusing
Pain and inflamation - I wonder if people really understood what the question means?
Ever taught in the curriculum - the question reads as if it is asking the student if they themselves did the teaching
and the same comment for Ever taught stewardship
138 - what does 'over prescribing mean'. Does it mean giving antibiotics when they are not needed? or does it refer to just the increasing use of antibiotics, even if they are clinically needed?
140 - a long duration is one driver, but difficult to determine if it is one of the main causes
141 should be 'Other' and not 'Others'
143 - incomplete sentence
136-146 There is no logical reason for why some of the data has been taken from the table and summarised, yet some data is excluded. Why have just a few been selected?
Table 3. 'Not removing the source'...this question is vague and can be interpreted in different ways.
Table 3 - the last question is misleading and not really relevant here.
150 delete the point about 'encouragingly' and leave these remarks for the Discussion
159 - don't use the word 'use' more than once
164 do you mean Stanford.
What about referring to local guidelines - American guidelines may not be applicable to Pakistan at times
Table 5. First question is confusing. Does it mean empirically or based on microbiology?
Table 6. Do these questions apply to the student carrying out the educational activities themselves; or if it is whether they think that a health professional is receiving the education from another source?
Formal teaching? confusing, what is this? does it mean delivered as part of the programme curriculum?
The questions about livestock are not relevant for any of the questionnaire. I understand the relationships with One Health, but the focus of this study has been on people
191-198 - the adjusted analyses needs further explanation. Was this based on hypotheses at the start of the study, or did you simply carry out lots of statistical tests - the latter is not correct approach.
Table 7. I don't understand the analysis. What and why was a mean rank used?
211 - Need to explain why the study is important in that it includes three student programmes all at once. So what?
214 - I would be far more cautious about using the term most? Even if it is 80%, that means that one in five people are incorrect
218 This sentence doesn't make sense.
219-223 This does not make sense. Nor is the point correct - all HCPs need to have the knowledge and correct behaviours
232, again be careeful of terms like 'majority'.
236 - can you add some data
240-243 the sentence is too long. It also doesn't make sense. What is the point that is being made? Why is information from pharma companies an issue?
246 - it isn't just international guidelines - it needs to be international and local guidelines.
264 - what type of bias and what impact on the findings
It is strange that the section on Materials and Methods, comes after the results. Normally it comes after the introduction.
274-277 - break down this sentence. It is too long.
275 the point about convenience sampling is explained later, so not needed here.
287 - you need to give a clear explanation regarding the sampling methods. Also add information about the number of enrolled students on each of the institutions programmes
291 you have repeated 'previously'
291 give short explanation about who the study was conducted on before.
293 how was validity measured
321 - assume? can you not find this out, even if roughly.
323 - the point about 50% yield does not make sense
326 - this should include non-responders
332 - how did you determine the preferences of the participants. Why didn't use the registration lists from each of the institutions?
334 how did you direct the social media postings to students at the universities? Other methods could have been used as well, such as contacting student societies
336 - surely the point about the online link and questions is irrelevant for an online questionnaire
352 - did you control for potential confounders? Did you have a hypothesis tests before the analysis was then conducted or was it all a-priori?
379 Conclusions. I think the main point of this study is to highlight that many HCP students do not graduate with high levels of knowledege and expertise to help them be competent. therefore it must be incorporated in the teaching curriculum at local and national level and part of the examination process
Author Response
English language and style
(x) Moderate English changes required
Author comment: Thank you. We have updated the paper with the help of one of the co-authors who is a native English speaker with over 450 publications in peer-reviewed Journals. We hope this is now OK.
Comments and Suggestions for Authors
1) The investigators have completed a valuable study that in essence, indicates the lack of formal and affective education in the curriculum for student HCPs in universities in Pakistan. Generally, the findings are consistent with other studies, in that knowledge and understanding has important gaps, and that understanding action necessary with regards to appropriate prescribing and stewardship, is lacking too. The study is a basic cross-sectional study. I find the sections on sampling, sampling source, to be confusing, as indicated in my comments. Similarly, the methods for multivariate analysis, adjusting for potential confounders, and primary stated hypotheses, are lacking and/or missing from the study execution. Some of the variables reported did not appear to be used in the analysis, such as income and parental profession - nor was there an explanation as to why this data was collected.
Author response: Thank you. We have modified the information regarding sampling. We have also performed regression analysis. The analysis regarding income and parental profession has also been performed and commented on (these factors were considered in the earlier studies that we built on). Hopefully, this is now acceptable.
2) The questionnaire, or at least the representation of the question as written in the tables, has quite a few confusing/ambiguous questions. Whilst the investigators claim to have used a previous questionnaire, that does not mean it is valid and reliable. There is a statement that internal validity was tested, but no evidence of how, nor the specific outcome. The quality of the questionnaire is my biggest concern with then making generalizations from the results.
Author response: Thank you. The questionnaire used in this study was first piloted with a small group of potential participants to add robustness to this. Besides, internal consistency was also measured by using Cronbach’s alpha which was higher than 7. This value indicates that our questionnaire is good enough to be used for this study. In addition, the pilot study participants fully understood the questions, and have now updated the methodology section accordingly. We have used this approach extensively in previously published studies, and hope this is now acceptable.
3) I found a lot of grammatical errors and sentences that did not make sense. These are indicated below. Some points needed to have more explanation.
Author response. Thank you. We have updated the paper with the help of one of the co-authors who is a native English speaker with over 450 publications in peer-reviewed Journals. We hope this is now OK.
4) The abstract is poorly written and needs to be clearly aligned with the highlights from the results.
Author response: Thank you – now updated.
5) 61 LMIC needs to be in brackets
Author response: Thank you. We have updated it.
6) 63-67 needs some data - it's all a bit vague.
Author response: Thank you – this is based on a synthesis of these papers. We have though modified the text and hope this is now acceptable.
7) 67-71 - need to either make a brief statement, and nothing else, about added burden on an already weak healthcare service; or you need to expand and explain the points.
Author response Thank you. We have modified these lines and hopefully now acceptable.
8) 72 - give data on the level of inappropriate use of antibiotics
Author response: Thank you for the comment. As you are no doubt aware, there are no exact data/figured to fully describe the level of inappropriate antibiotic use across indications in LMICs. Most of the studies report that majority of the antibiotics use in primary care health facilities in LMICs is typically inappropriate, i.e. extensive prescribing for URTIs and other predominantly self-limiting conditions. However, it is difficult to put an exact figure on this with very limited point of care testing in LMICs. We should know as we have published extensively in this area across LMICs (some references have now been added to help explain this further for those interested). We have made mention to the NAPs which further document these concerns – and potential ways forward, and hope this is now acceptable.
9) 76 - give data
Author response: Thank you. We have provided data of bacterial or fungal co-infections based on systematic analyses that we and others have performed in this area. We hope this clarifies the situation.
10) 81 - why an appreciable problem globally? Explain
Author response: Thank you. We have modified the sentence.
11) 87 - 'these' instead of 'this'
Author response: Thank you. We have updated the sentence.
12) 88 - I would make the point somewhere about the regulation and over-the-counter purchasing
Author response Thank you. We have updated the sentence based on research we have previously conducted in this area, and hope this is now OK.
13) 95 - give examples of the challenges
Author response: Thank you. We have mentioned the challenges/difficulties to implement national action plan of Pakistan against AMR.
14) 51 - I don't think the role of HCPs is limited to LMIC - it is a global need
Author response: Thank you. We agreeing with this. However, the scale of problem is appreciable in LMICs (some of the co-authors have extensive experience researching and publishing on the management of patients across sectors in both LMICs and high-income countries and can readily agree with this). We hope this is now OK.
15) 100 - give an example of what they don't know. Also is this comparable to other countries or unique to Pakistan?
Author response: Thank you. We have mentioned that the HCWs did possess insufficient awareness and training about antibiotic use, AMR and ASPs. Moreover, this problem is common in many LMICs. We hope this is now OK.
16) 101 - This point has already been made. Delete sentence.
Author response: Thank you. We have updated it as per your comment.
17) 105 - I would start a new paragraph that makes the point that the basis for appropriate clinical care starts from the beginning of their training. Then add that this is a concern, and give a few more examples - including a recent study by David McMaster et. al., doi: 10.1093/jacamr/dlaa096. eCollection 2020 Dec.nd also work by Molly Courtenay
Author response: Thank you. We have updated the manuscript as per your comment by incorporating the references suggested by you and others. We hope this is now OK.
18) 110 - how could you 'invite' the participants if you had simply used social media as a way to try and recruit anyone seeing the social media post?
Author response: Thank you. We recruited study participants with the assistance of administration/societies of the students. These societies had different social media groups/connections. Moreover, the individual participants were approached by the investigators with the assistance of institutional administration that allow us to collect data from them. This updated information has now been incorporated in updated version of manuscript, and we hope this is now OK.
19) and how could you determine the number initially invited?
Author response: Thank you. Once we have shared the e-questionnaire, we contacted study participants twice (with gap of three days) – as indicated in the manuscript. We hope this clarifies the situation.
20) 117 - how did you define 'rural area'?
Author response: Thank you. As per the Government of Pakistan, every citizen issued by National Identity card that states their status of residency. We hope this clarifies the situation.
21) 117 29.5% is not a quarter
Author response: Thank you. We have updated it as per your comment.
22) 117 the point about ASP training should come after all the demographics were explored.
Author response: Now moved
23) 119 I don't understand why you collected data on parent profession.
Author response: Thank you. We collected this information from the study participants to see if there is any association between profession of parents with the understanding of their antibiotic use, AMR, etc., as parents of professionals may have discussed such issues in general in the household as children are growing up. In addition, we included this data in our previous studies. In fact, as seen by new Table 8 – no difference. We hope this is now OK.
24) 125 - 'the majority....' this is subjective, and should be in the discussion. Results should stick to the facts.
Author response. Thank you – now amended
25) Table 2 - skipping doses - I suspect it needs to be more than one or two doses.
Author response: Thank you – you are correct. It means that if one, two or more than two doses, etc., of antibiotics were missed, which may lead to AMR (as understood by the participants). This question was reverse coded and the participants need to select ‘No’ (No was the right answer here). I hope this will be acceptable now.
26) Cross-resistance - the question is confusing
Author response: Thank you – this is defined in the Table. We hope this is OK.
27) Pain and inflammation - I wonder if people really understood what the question means?
Author response: Yes we believe so as this Questionnaire was tested in a pilot with students and there was a good understanding score before full implementation. We have made this point in the revised paper, and hope this is acceptable to you
28) Ever taught in the curriculum - the question reads as if it is asking the student if they themselves did the teaching and the same comment for Ever taught stewardship.
Author response: Thank you – the students in the pilot study interpreted this as ever being taught AMR in their current curriculum – so no real concerns in reality. We hope this is now clarified.
29) 138 - what does 'over prescribing mean'. Does it mean giving antibiotics when they are not needed? or does it refer to just the increasing use of antibiotics, even if they are clinically needed?
Author response: Thank you – these students and others know this means prescribing antibiotics when not needed such as for self-limiting conditions such as URTIs as well as over prescribing in patients with COVID-19. We hope this clarifies the situation
30) 140 - a long duration is one driver, but difficult to determine if it is one of the main causes.
Author response: Thank you – however we beg to differ. For instance, in hospitals in LMICs including those in Africa/ Asia, there is typically extensive prescribing of antibiotics beyond one day post-operatively. This drives up AMR rates without improving patient outcomes – so is a ley area for ASPs in hospitals (discussed extensively in Mwita JC et al. Key Issues Surrounding Appropriate Antibiotic Use for Prevention of Surgical Site Infections in Low- and Middle-Income Countries: A Narrative Review and the Implications. Int J Gen Med. 2021;14:515-30 and Saleem Z et al. Ongoing Efforts to Improve Antimicrobial Utilization in Hospitals among African Countries and Implications for the Future. Antibiotics. 2022;11:1824). We hope this clarifies the situation.
31) 141 should be 'Other' and not 'Others'
Author response: Thank you. We have updated it.
32) 143 - incomplete sentence
Author response: Thank you. We have updated it.
33) 136-146 There is no logical reason for why some of the data has been taken from the table and summarised, yet some data is excluded. Why have just a few been selected?
Author response: Thank you – in line with an appreciable number of other publications we have been involved with – we just concentrated on key areas in the narrative. We hope this clarifies the situation.
34) Table 3. 'Not removing the source'...this question is vague and can be interpreted in different ways.
Author response: Thank you for this. However, our students in the pilot and beyond knew that for instance an appreciable number of hospital-acquired infections are caused by indwelling devices. Consequently, removing these where pertinent reduces the chances of HAI, etc. We hope this clarifies the situation.
35) Table 3 - the last question is misleading and not really relevant here.
Author response: Thank you. However we beg to differ. In LMICs such as Pakistan, there is no real control over pharmaceutical company activities (as seen for instance in higher-income countries with voluntary agreements). Alongside this in e.g. hospitals limited number of active DTCs to counteract pharmaceutical company activities. We and others have published on these concerns – especially surrounding the lack of ADR information in company leaflets, and the appreciable influence of pharma company activities on prescribing, etc., and have included some of this information, etc., in the Discussion along with pertinent references. We hope this clarifies the situation.
36) 150 delete the point about 'encouragingly' and leave these remarks for the Discussion.
Author response: Thank you. We have updated this in line with earlier suggestions, and hope this is now acceptable.
37) 159 - don't use the word 'use' more than once
Author response: Thank you now changed
38) 164 do you mean Stanford.
Author response: Thank you. It refers to SANFORD GUIDE for ‘Antimicrobial Stewardship’. We hope this clarifies the matter.
37) What about referring to local guidelines - American guidelines may not be applicable to Pakistan at times
Author response: Thank you. As you may be aware in a number of LMICs including Pakistan, antimicrobial guidelines are not readily available. Consequently, most of the HCWs use international guidelines if at all. This may change now that the WHO AWaRe guidance is published and easily available, building on the AWaRe classification of antibiotics, and we will be following this up in future research projects (we have now mention this in the updated paper). We hope this is clear.
38) Table 5. First question is confusing. Does it mean empirically or based on microbiology?
Author response: Thank you. It means we are referring to both empiric or targeted therapy based on culture and sensitivity testing, which typically does not take place in LMICs such as Pakistan. This is a key area for the future to improve antimicrobial use – especially around Watch/ Reserve antibiotics (AWaRe classification) – and we will be following this up in the future. Hopefully, this clarifies the situation.
39) Table 6. Do these questions apply to the student carrying out the educational activities themselves; or if it is whether they think that a health professional is receiving the education from another source?.
Author response: Thank you. These questions refer to potential ways to address AMR. In this respect, HCW education is key to reducing high rates of inappropriate prescribing and dispensing of antibiotics in Pakistan in the future and thereby reducing AMR. We hope this is now OK.
40) Formal teaching? confusing, what is this? does it mean delivered as part of the programme curriculum?
Author response: Thank you. Here formal education means the curriculum of the respective cadre including medical, pharmacy or nursing school curriculum, which was understood by the students in the pilot study. We hope this is now OK.
41) The questions about livestock are not relevant for any of the questionnaire. I understand the relationships with One Health, but the focus of this study has been on people.
Author response: Thank you. However, we again beg to differ especially among LMICs. This question merely evaluates their perception to address AMR besides enhancing appropriate antibiotic use in humans. This is important in LMICs such as Pakistan with high rates of self-purchasing of antibiotics through pharmacy/ drug stores including for animals. For instance, in South Africa there was appreciable use of ivermectin (anti-parasitic in animals) purchased from pharmacy/ drug stores to treat COVID-19 without any evidence and in Zambia high rates of dispensing poultry antibiotics without a prescription which is a concern as this also drives up AMR (we have published on both in Antibiotics). Finally, there are now strict regulations regarding the dispensing of colistin in South Africa due to concerns. Consequently, we have kept this question in. We hope this clarifies the situation.
42) 191-198 - the adjusted analyses needs further explanation. Was this based on hypotheses at the start of the study, or did you simply carry out lots of statistical tests - the latter is not correct approach.
Author response: Thank you. We have assessed the association between the demographic characteristics of the study participants with the knowledge of antibiotics, awareness about causes of AMR, perceptions of AMR, pre-paredness towards ASPs, and approaches to tackle AMR. This builds on earlier studies. We hope this is acceptable.
43) Table 7. I don't understand the analysis. What and why was a mean rank used?
Author response: Thank you. We used mean rank for the results of kruskal Wallis and Mann Whitney u tests. We hope this clarifies the situation.
44) 211 - Need to explain why the study is important in that it includes three student programmes all at once. So what?
Author response: Thank you – we have now said why this is important as typically multidisciplinary teams are being organized in hospitals as part of ASPs to improve future antimicrobial use. We hope this is now acceptable.
45) 214 - I would be far more cautious about using the term most? Even if it is 80%, that means that one in five people are incorrect
Author response: Thank you – now revised.
46) 218 This sentence doesn't make sense.
Author response: Thank you – now changed
47) 219-223 This does not make sense. Nor is the point correct - all HCPs need to have the knowledge and correct behaviour
Author response: Thank you – now updated. We hope this is now OK.
48) 232, again be careful of terms like 'majority'.
Author response: Thank you – now changed
49) 236 - can you add some data
Author response: Thank you – we have expanded the data on Rwanda.
50) 240-243 the sentence is too long. It also doesn't make sense. What is the point that is being made? Why is information from pharma companies an issue?
Author response: Thank you. We have expanded on this in line with earlier comments and provided additional information. We hope this is now OK.
51) 246 - it isn't just international guidelines - it needs to be international and local guidelines.
Author response: Thank you – as mentioned earlier local guidelines are not available/developed in Pakistan – although the new WHO AWaRe guidance is a good step forward. We hope this clarifies the situation.
52) 264 - what type of bias and what impact on the findings
Author response: Bias in terms of some of the %s in the findings because our sampling was convenient rather than random. It is normal to write this kind of statement with this methodology. We hope this clarifies the situation.
53) It is strange that the section on Materials and Methods, comes after the results. Normally it comes after the introduction.
Author response: Thank you. It is the journal requirement/style. We hope this clarifies the situation.
54) 274-277 - break down this sentence. It is too long.
Author response: Thank you. We have updated it now.
55) 275 the point about convenience sampling is explained later, so not needed here.
Author response: Thank you. We have updated it now.
56) 287 - you need to give a clear explanation regarding the sampling methods. Also add information about the number of enrolled students on each of the institution’s programmes
Author response: Thank you. We have incorporated sampling methods in ‘data collection procedure’ Moreover information about the total number of enrolled students in respective institutions have been incorporated in updated version of manuscript. We hope this is now acceptable.
57) 291 you have repeated 'previously'
Author response: Thank you. We have corrected it in updated version of manuscript.
58) 291 give short explanation about who the study was conducted on before.
Author response. Thank you. We have incorporated this information in updated version of manuscript. We hope this is now acceptable.
59) 293 how was validity measured
Author response: The validity was measured by measured by using Cronbach’s alpha. This was also mentioned in the updated version of manuscript. We hope this clarifies the situation.
60) 321 - assume? can you not find this out, even if roughly.
Author response: Thank you for the comment. There was no centralized data base available in Pakistan to describe precise numbers of these final year graduates. The authors themselves collected this information from individual institutional and described it in the concerned sections – leading to these comments. We hope this clarifies the situation.
61) 323 - the point about 50% yield does not make sense
Author response: Thank you. We have modified it to provide comprehensive meaning. We hope this is now OK.
62) 326 - this should include non-responders
Author response: Thank you. We have updated it.
63) 332 - how did you determine the preferences of the participants. Why didn't use the registration lists from each of the institutions?
Author response: Thank you for the comment. We requested this information from the administration/societies of the concerned institutes and they assisted us in determining participants preferences. For example, nursing students were approached through individual institutional email as their institutions didn’t allow other social sites to collect data for research purposes. We hope this clarifies the situation.
64) 334 how did you direct the social media postings to students at the universities? Other methods could have been used as well, such as contacting student societies.
Author response: Thank you. We approached the study participants by requesting the administration/societies of institute to help us in contacting their students through social media sites. We hope this clarifies the situation.
65) 336 - surely the point about the online link and questions is irrelevant for an online questionnaire
Author response: Thank you. In this way, potentially more students could take part in the questionnaire. We hope this clarifies the situation.
66) 352 - did you control for potential confounders? Did you have a hypothesis test before the analysis was then conducted or was it all a-priori?.
Author response. Thank you for the comment. We did not have any prior hypothesis to test as this was very much a situational analysis among these 3 student HCP groups – who are the future to try and reduce high AMR rates across Pakistan. We have tried to address key issues by a new multivariate analysis. Hopefully, this is now OK.
67) 379 Conclusions. I think the main point of this study is to highlight that many HCP students do not graduate with high levels of knowledge and expertise to help them be competent. therefore, it must be incorporated in the teaching curriculum at local and national level and part of the examination process.
Author response: Thank you for the suggestion. The conclusion section has now been updated, and we hope this is now acceptable.
Reviewer 3 Report
A cross-sectional survey evaluating knowledge of antibiotic use, antimicrobial resistance and antimicrobial stewardship among future healthcare professionals. This is generally well-written and the survey items appear to have good validity with a sound basis in existing research. However, there are some issues, particularly with the analysis of the association of demographic variables with the outcomes of interest.
The analysis as it stands is not particularly informative. This tells us which of the factors are associated with the outcomes, but not the direction of the association, nor the effect of the specific subcategories, i.e. do males have a positive association with knowledge relative to females? This seems more like the stage prior to a regression analysis to help decide which covariates to include in the model. It would also be better to run a multivariable analysis so you can look at the effect of these covariates in the presence of one another. However, for this, you would need to rethink the analysis plan and clearly state your exposure(s) of interest. Related to this, there needs to be some further rationale to explain the choice of demographic characteristics listed in Table 1. Is there evidence in the literature that family income, student type or institute type might have any bearing on one’s understanding of AMS? Perhaps one of these characteristics could provide the basis for a solid hypothesis which could then facilitate a clear analysis plan.
You’ve stated you used “a convenient sampling technique”, but not provided any detail about what you did to promote the study and recruit your participants. The limitations in the discussion section need further detail. It is insufficient to say the findings are robust because you “believe” that they are.
Author Response
Comments and Suggestions for Authors
A cross-sectional survey evaluating knowledge of antibiotic use, antimicrobial resistance and antimicrobial stewardship among future healthcare professionals. This is generally well-written and the survey items appear to have good validity with a sound basis in existing research. However, there are some issues, particularly with the analysis of the association of demographic variables with the outcomes of interest.
Author response: Thank you for your comment. we will address all your comments.
1) The analysis as it stands is not particularly informative. This tells us which of the factors are associated with the outcomes, but not the direction of the association, nor the effect of the specific subcategories, i.e. do males have a positive association with knowledge relative to females? This seems more like the stage prior to a regression analysis to help decide which covariates to include in the model. It would also be better to run a multivariable analysis so you can look at the effect of these covariates in the presence of one another. However, for this, you would need to rethink the analysis plan and clearly state your exposure(s) of interest. Related to this, there needs to be some further rationale to explain the choice of demographic characteristics listed in Table 1. Is there evidence in the literature that family income, student type or institute type might have any bearing on one’s understanding of AMS? Perhaps one of these characteristics could provide the basis for a solid hypothesis which could then facilitate a clear analysis plan.
Author response: Thank you for the comment. We have updated analysis section and also included the factors associated with the better antibiotic use knowledge, perception and causes of AMR, preparedness towards hospital-based ASP and ways to address AMR with an updated analysis. Issues such as family income, student type and institution type have featured in our previous studies with medical and pharmacy students separately – and also been seen in other studies in LMICs (which we now reference in the Discussion). Moreover, our new regression analysis (new Table 8) has shown that type of students, and type of institutions has a significant (p<0.05) influence about their ASP training. We know based on our extensive activities across multiple LMICs that identifying these issues is very important to train the next generation of HCPs given ever rising AMR rates in Pakistan and beyond and the need to attain the goals in the NAP for AMR in each country. As mentioned, we developed the questionnaire including the demographics based on our previous studies in this area (referenced), and updated following the pilot, with this study very much a situational analysis containing all 3 critical HCP student groups in Pakistan (rather than just medical and pharmacy students separately). As such, no specific pre-determined hypotheses to test. We hope this is now acceptable.
2) You’ve stated you used “a convenient sampling technique”, but not provided any detail about what you did to promote the study and recruit your participants. The limitations in the discussion section need further detail. It is insufficient to say the findings are robust because you “believe” that they are.
Author response: Thank you. We have further illustrated the method of participants recruitment in our survey in the updated paper. We have also updated the methodology section to show further details of the pilot study and the validity, etc. This is why we believe the findings are robust. However, we have also updated the limitations section of the study. We hope this is now OK.
Reviewer 4 Report
In the present manuscript, the authors assessed knowledge about antibiotic use, antimicrobial resistance (AMR), and readiness for antimicrobial stewardship programs (ASPs) among 1251 final year medical, pharmacy and nursing students from 23 public and private educational institutions in Punjab Province, Pakistan. In general, the manuscript is well written and the study design is appropriate. However, the study could have been more interesting if the authors had also compared the results between the different categories of students.
A major point to be addressed:
Lines 193-198: Authors should discuss in the corresponding section the possible causes that could explain the significant associations found between demographic variables of the students with their knowledge of antibiotics and causes of AMR, perception towards AMR and preparedness of ASPs, and approaches used to tackle AMR.
Some typos:
Lines 1-4: Please delete the hyphens and change the semicolon to colon.
Line 38: “though” or thought?
Line 62: LMICs in parentheses.
Line 194: Please delete “it’s”.
Line 245: Please delete “though”.
Author Response
Comments and Suggestions for Authors
1) In the present manuscript, the authors assessed knowledge about antibiotic use, antimicrobial resistance (AMR), and readiness for antimicrobial stewardship programs (ASPs) among 1251 final year medical, pharmacy and nursing students from 23 public and private educational institutions in Punjab Province, Pakistan. In general, the manuscript is well written and the study design is appropriate. However, the study could have been more interesting if the authors had also compared the results between the different categories of students.
Author response: Thank you for review and comments. We have tried to address your comments and suggestions.
2) A major point to be addressed: Lines 193-198: Authors should discuss in the corresponding section the possible causes that could explain the significant associations found between demographic variables of the students with their knowledge of antibiotics and causes of AMR, perception towards AMR and preparedness of ASPs, and approaches used to tackle AMR.
Author response: Thank you. We have made some suggestions based on this and other research projects (now cited). However further research is needed to be able to fully answer this. Consequently, we have added this as a limitation. We hope this is acceptable.
3) Some typos:
- i) Lines 1-4: Please delete the hyphens and change the semicolon to colon.
Author response: Thank you – now implemented.
- ii) Line 38: “though” or thought?
Author response: Thank you – this is though
iii) Line 62: LMICs in parentheses.
Author response: Thank you – now implemented.
- iv) Line 194: Please delete “it’s”.
Author response: Thank you – now implemented.
- v) Line 245: Please delete “though”
Author response: Thank you – now implemented.
Round 2
Reviewer 1 Report
Thank you for the responses. I have no further comment.
Author Response
Thank you for the responses. I have no further comment.
Author comment: Thank you for this and your help with updating the paper – appreciated!

Reviewer 2 Report
Hi, the authors' have considered my previous report and made many changes throughout. Whilst this is likely to be the first report on this for Pakistan, the results are of little surprise, and in keeping with reports from other LMIC and elsewhere. It has some basic value, and does help to highlight the need for education across professional education/training. There is a lot of data/tables presented, and I don't think these all need to be in the main text. Consider having online appendices.
I don't fully agree with the challenge to the issues about the quality of the questionnaire. However, some aspects are evidently likely to be true, given other background literature.
I'm not averse to this paper being published. It does require a final spell check and some minor attention to grammar and so forth.
Regards
Roger
Author Response
Comments and Suggestions for Authors
1) Hi, the authors' have considered my previous report and made many changes throughout. Whilst this is likely to be the first report on this for Pakistan, the results are of little surprise, and in keeping with reports from other LMIC and elsewhere. It has some basic value, and does help to highlight the need for education across professional education/training. There is a lot of data/tables presented, and I don't think these all need to be in the main text. Consider having online appendices.
Author comment: Thank you for your kind comments. We appreciated your previous input. We have now included some Figures and Tables in the Appendix (Supplementary Material) as well as removed one Table. We hope this is now OK.
2) I don't fully agree with the challenge to the issues about the quality of the questionnaire. However, some aspects are evidently likely to be true, given other background literature.
Author comment: Thank you for this.
3) I'm not averse to this paper being published. It does require a final spell check and some minor attention to grammar and so forth.
Author comment: Thank you for this. We have now further refined the paper and hope this is now acceptable.
Reviewer 3 Report
Thank you for responding to my comments. The paper is much improved. Before I can recommend acceptance, there are still some remaining issues to address. I appreciate the time and effort that has gone into advancing the analysis, however, it needs some further consideration. You may want to seek the guidance of a statistician to check your interpretation prior to resubmission. Furthermore, the value of the results to your overall findings needs greater emphasis and should be weaved into your overarching narrative thread. I think the difficulty of only including this as a response to a reviewer comment is that you have not taken the time to carefully consider how and why you are conducting this analysis and what value this is adding to the aims and objectives of your study and, more broadly, to the literature on the topic.
- Please include the most pertinent ORs and corresponding 95% CIs from your logistic regression analysis into the results section of your abstract.
- It is inaccurate to describe the results from your analysis in Table 7 as showing “Female study participants, family income <25000 PKR, and resident in rural areas were associated with better antibiotic knowledge, causes and perception of AMR.” This analysis can only tell you if there are statistical differences between your groups, it cannot identify associations or the directions of those associations. That is what your logistic regression is for.
- You do not make clear whether you have conducted univariate or multivariate regression analysis to produce your ORs in Table 8. This should be stated in the methods and the results table. It would be ideal if you conducted both and presented the results from each. You also need to highlight the key findings from this analysis in the supporting text – as it stands you only provide the most cursory summarising sentence.
- You need to discuss the findings from the regression analysis in the discussion section of the paper.
- I think there remains a lack of detail on how the sampling was conducted. Yes, I understand it was convenience sampling, but how were institutions identified and contacted? Did you contact all relevant institutions and wait to see who responded (not really convenience sampling)? Or was this based on pre-existing contacts/relationships that the authors had with these places? Was any effort made to include particular traits, i.e. public/private, geographical location? If so, would you consider this to involve an element of purposive sampling?
- The limitations section of the discussion is better but needs further work. Please explain in what way precisely your sampling approach might introduce bias to your findings. You also need to consider the typical limitations of causal inference through regression modelling and how these may have biased your findings. Most important is the possibility of omitted variable bias, but there is also multicollinearity, missing data and measurement error to be considered.
Author Response
(x) English language and style are fine/minor spell check required
( ) I don't feel qualified to judge about the English language and style
Author Response: Thank you – we have now been through the manuscript again and undertaken further English language checks, etc. We hope this is now OK.
Comments and Suggestions for Authors
1) Thank you for responding to my comments. The paper is much improved. Before I can recommend acceptance, there are still some remaining issues to address. I appreciate the time and effort that has gone into advancing the analysis, however, it needs some further consideration. You may want to seek the guidance of a statistician to check your interpretation prior to resubmission. Furthermore, the value of the results to your overall findings needs greater emphasis and should be weaved into your overarching narrative thread. I think the difficulty of only including this as a response to a reviewer comment is that you have not taken the time to carefully consider how and why you are conducting this analysis and what value this is adding to the aims and objectives of your study and, more broadly, to the literature on the topic.
Author Response: Thank you for this. We have now improved the paper. This includes updating the aims of our study and how the findings can be used. In addition, adding in more details to the Discussion and Conclusion, and hope this is now acceptable.
2) Please include the most pertinent ORs and corresponding 95% CIs from your logistic regression analysis into the results section of your abstract.
Author Response: Thank you. We have updated the abstract accordingly, and hope this is now OK.
3) It is inaccurate to describe the results from your analysis in Table 7 as showing “Female study participants, family income <25000 PKR, and resident in rural areas were associated with better antibiotic knowledge, causes and perception of AMR.” This analysis can only tell you if there are statistical differences between your groups, it cannot identify associations or the directions of those associations. That is what your logistic regression is for.
Author Response: Thank you. We have modified the interpretation of the Table 7 and removed the above sentence. We hope this is now OK.
4) You do not make clear whether you have conducted univariate or multivariate regression analysis to produce your ORs in Table 8. This should be stated in the methods and the results table. It would be ideal if you conducted both and presented the results from each. You also need to highlight the key findings from this analysis in the supporting text – as it stands you only provide the most cursory summarising sentence.
Author Response: Thank you. We performed a multivariate regression analysis. This information has been added in the revised version and we have removed old Table 8 to just concentrate on the key findings (and in line with the comments from another reviewer). We hope this is now OK.
5) You need to discuss the findings from the regression analysis in the discussion section of the paper.
Author Response: Thank you. We have now added in more details in the Discussion together with an additional reference and hope this is now OK.
6) I think there remains a lack of detail on how the sampling was conducted. Yes, I understand it was convenience sampling, but how were institutions identified and contacted? Did you contact all relevant institutions and wait to see who responded (not really convenience sampling)? Or was this based on pre-existing contacts/relationships that the authors had with these places? Was any effort made to include particular traits, i.e. public/private, geographical location? If so, would you consider this to involve an element of purposive sampling?
Author Response Thank you for the comment. We contacted all public and private sector medical, pharmacy and nursing institutions in the Punjab Province for participation in the study. However, we could only include those institutions whose administration/ societies allowed/supported us in the data collection procedure. We have mentioned this in the revised version of the manuscript. Moreover, we have included study participants from seven divisions of the province (Out of a total of ten) in the revised paper as well as provided greater details of the Institutions involved in the Supplementary Material. We hope this is now OK.
7) The limitations section of the discussion is better but needs further work. Please explain in what way precisely your sampling approach might introduce bias to your findings. You also need to consider the typical limitations of causal inference through regression modelling and how these may have biased your findings. Most important is the possibility of omitted variable bias, but there is also multicollinearity, missing data and measurement error to be considered.
Author Response. Thank you. We have modified the limitations section accordingly. We hope this is now acceptable.
Reviewer 4 Report
The authors have sufficiently improved the manuscript to warrant publication in Antibiotics.
Author Response
Comments and Suggestions for Authors
The authors have sufficiently improved the manuscript to warrant publication in Antibiotics.
Author comment: Thank you for this and your help with updating the paper – appreciated!